# Reversible flexoelectric domain engineering at the nanoscale in van der Waals ferroelectrics

Heng Liu[1,2,3,6], Qinglin Lai[1,2,3,6], Jun Fu[1,2,3], Shijie Zhang[4,5], Zhaoming Fu[4,5] & Hualing Zeng[1,2,3] ✉

The universal flexoelectric effect in solids provides a mechanical pathway for controlling electric polarization in ultrathin ferroelectrics, eliminating potential material breakdown from a giant electric field at the nanoscale. One challenge of this approach is arbitrary implementation, which is strongly hindered by one-way switching capability. Here, utilizing the innate flexibility of van der Waals materials, we demonstrate that ferroelectric polarization and domain structures can be mechanically, reversibly, and arbitrarily switched in two-dimensional $CuInP_2S_6$ via the nano-tip imprinting technique. The bidirectional flexoelectric control is attributed to the extended tip-induced deformation in two-dimensional systems with innate flexibility at the atomic scale. By employing an elastic substrate, artificial ferroelectric nanodomains with lateral sizes as small as ~80 nm are noninvasively generated in an area of 1 μm$^2$, equal to a density of 31.4 Gbit/in$^2$. Our results highlight the potential applications of van der Waals ferroelectrics in data storage and flexoelectronics.

The emergence of room-temperature ultrathin ferroelectrics, including hafnium oxide[1], perovskite oxides[2], and van der Waals (vdW) materials[3–6], vacillates the long-recognized critical size effect of ferroelectricity[7] and offers the potential to develop miniaturized ferroelectric devices[8,9] that can be integrated into modern semiconductor technology. The first step in validating the device's practicability is to demonstrate the effective control of electric polarizations and reliable engineering of domains in these two-dimensional (2D) ferroelectrics with high fidelity. Applying an external voltage to switch ferroelectric polarizations normally seems to be an optimal method[10]. Nevertheless, due to the atomic thickness of 2D materials, the giant electric field at the nanoscale results in inevitable functional problems, including charge injections and material breakdowns[11–14]. In this context, exploring alternative strategies for lowering or even abandoning electric poling is highly important in both fundamental and technology for 2D ferroelectrics[15–20].

Recent progress has shown that flexoelectricity can be utilized as a voltage-free method for manipulating ferroelectric properties via the coupling of strain gradients and electric polarizations[11,13,15,16,21]. This effect benefits the universality of all materials[22–27] and is anticipated to be enhanced in low-dimensional systems due to the inverse scale effect of strain gradients on material size[28]. However, to date, the widespread implementation of flexoelectric control in ferroelectrics is far less common than expected in the realm. A possible explanation is that the mechanical switching pathway for electric polarizations is usually limited to a single or fixed direction. For example, in bulk perovskite oxides, nanoscale strain gradients can be generated externally by pressing an atomic force microscope (AFM) tip or inherently by atomic lattice mismatch[15,29,30], where flexoelectric control is unidirectional. In vdW ferroelectrics, effective flexoelectric modulation can be realized by local sharp topographic features from the patterned substrate, yet this approach results in fixed domain structures[11,16]. To that end, there

[1]International Center for Quantum Design of Functional Materials (ICQD), Hefei National Research Center for Physical Sciences at the Microscale, University of Science and Technology of China, Hefei 230026, China. [2]CAS Key Laboratory of Strongly Coupled Quantum Matter Physics, Department of Physics, University of Science and Technology of China, Hefei, Anhui 230026, China. [3]Hefei National Laboratory, University of Science and Technology of China, Hefei 230088, China. [4]College of Physics and Electronic Information, Yunnan Normal University, Kunming 650500, China. [5]Yunnan Key Laboratory of Opto-Electronic Information Technology, Kunming 650500, China. [6]These authors contributed equally: Heng Liu, Qinglin Lai. ✉e-mail: hlzeng@ustc.edu.cn

is a need to improve the flawed flexoelectric control approach at the current stage.

In this work, we report the mechanical realization of arbitrary polarization reversal and artificial domain engineering in ultrathin ferroelectric $CuInP_2S_6$ (CIPS) via the tip imprinting technique. Under the pressure of an AFM tip at the nanoscale, we observed that the morphology of CIPS undergoes an extended deformation, as depicted by a Gauss curved face function in theory, which is distinct from the case of hard contact in bulk ferroelectrics[15,31]. The soft deformation is attributed to the innate flexibility of vdW materials and generates a bidirectional flexoelectric field that reverses the polarizations of CIPS in two-way, from upward to downward and vice versa. With this technique, nanodomains of different shapes, such as dots and rings, were selectively generated in pre-polarized samples. In particular, by applying different tip forces to reverse the downward ferroelectric domains, a transition of the tri-petal domain to the ring domain was observed. The emergence of tri-petal domains suggested the existence of in-plane anisotropic flexibility in CIPS. With a further increase in the flexoelectric field, the unique high-polarization state in the CIPS was realized. Finally, arbitrary flexoelectric engineering was achieved by employing an elastic substrate and noninvasively generated high-density ferroelectric nanodomains (31.4 Gbit/in²). Our findings suggest a universal route to mechanically modulate the polarizations in ultrathin ferroelectrics and highlight the potential of vdW ferroelectrics in memory applications.

## Results

### Strategy for two-way flexoelectric control

The key to flexoelectric control of polarization is creating sufficiently large strain gradients[28,32]. For instance, in conventional ferroelectric oxides, nanoscale strain gradients can be controllably induced by using the scanning probe of an AFM, as illustrated in Fig. 1a[15]. In this case, the tip is in hard contact with the ferroelectrics, causing

negligible sample deformation due to its rigid bulk nature. Nonetheless, flexoelectricity can still be triggered due to the uneven distribution of tip-induced pressure on the surface. Previous studies[33] have shown that a significant downward strain gradient emerges, enabling flexoelectric control of the upward polarization. While reversed upward strain gradients are identifiable when examining the cross-sectional distribution in the imprinting area (see the inset in Fig. 1a), they are confined within a spatial scale of one to two nanometers. This highly localized flexoelectric field fails to stabilize upward ferroelectric polarization, as constrained by the long-range Coulomb interactions between electric dipoles[34]. Thus, if the spatial limit can be removed, the tip-induced flexoelectric field can be bidirectionally utilized to arbitrarily switch the polarization.

Recent progress has shown remarkable flexibility and superelasticity in ultrathin films, including free-standing perovskite oxides[35,36] and vdW 2D materials[37,38]. Upon mechanical stimulus, the flexibility of these ultrathin materials has a significant influence on the strain distributions because of the substantial deformation of the samples[39], which holds the potential to overcome the issue of a single mechanical switching pathway in bulks. For example, when the same nanoscale tip imprinting technique is applied in ultrathin ferroelectrics, extended indentation beyond the contact area is expected (see Fig. 1b). Our first-principle calculations (detailed in Supplementary Note 1 and Fig. S1) reveal that the geometric deformation of 2D ferroelectrics along the cross section can be effectively depicted by a Gaussian curvature function:

$$z(x) = z_0 + Ae^{-\frac{(x-\mu)^2}{2\sigma^2}}, \tag{1}$$

where $z(x)$ represents the height changes of the surface morphology and $z_0$ and $A$ are constants. Considering the realistic tip diameter ($\sim 20$ nm) in an AFM, the soft deformation in 2D ferroelectrics can be numerically depicted with Eq. (1) at a large scale ($\sim 100$ nm), as shown

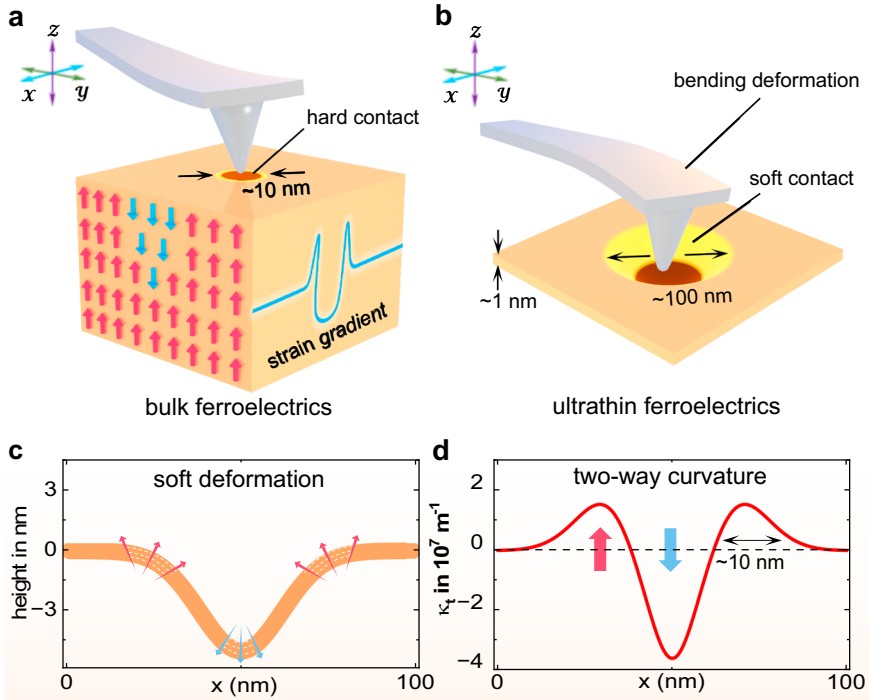

**Fig. 1 | Strategy for bidirectional modulation of ferroelectric polarization.**
**a**, **b** Schematic diagrams of the tip-induced flexoelectric control in conventional bulk ferroelectrics and ultrathin ferroelectrics. The red and blue arrows in the bulk ferroelectrics indicate the upward and downward polarizations, respectively. The inset shows the spatial distribution of the strain gradients induced by the hard

contact in the bulk material. **c** Cross-sectional view of the tip-induced soft deformation in ultrathin ferroelectrics. Arrows indicate the directions of curvatures. **d** The corresponding spatial distribution of the strain gradients for the soft deformation in (**c**). Arrows indicate the directions of the strain gradients.

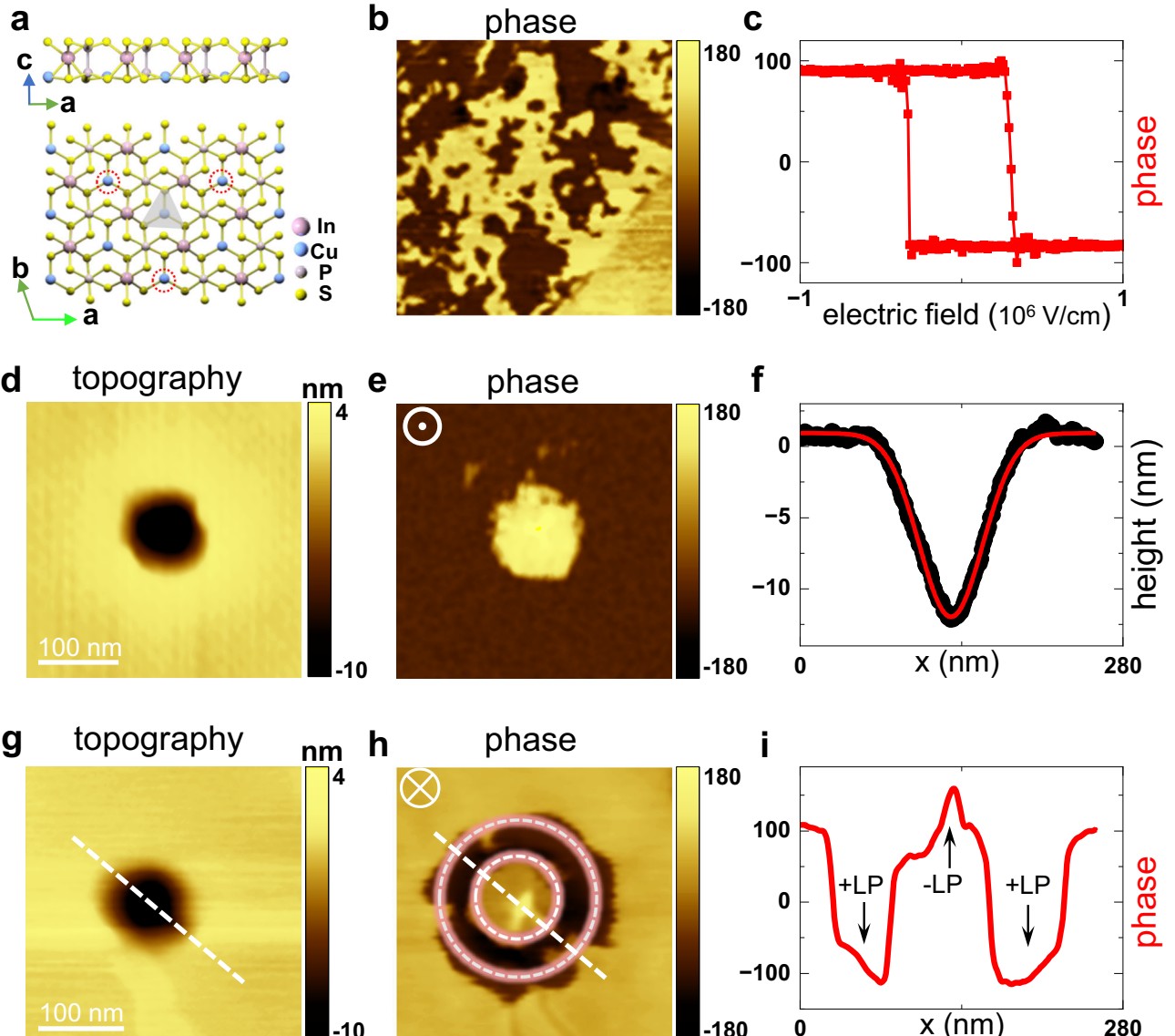

**Fig. 2 | Two-way flexoelectric control of ferroelectricity in CuInP₂S₆. a** Crystal structure of CIPS. **b**, **c** PFM phase image of a spontaneous ferroelectric domain and hysteresis loop from a 22 nm thick CIPS flake. **d**, **e** PFM height and phase images of an upwardly pre-polarized CIPS after 4 μN tip force imprinting. In the PFM phase image, the upward domain is marked with brown, and the downward domain is marked with yellow. **f** Selected line profiles of height derived from (**g**). The red line is the fitting of the Gaussian curvature function. **g**, **h** PFM height and phase images of a downwardly pre-polarized CIPS after 4 μN tip force imprinting. The ring-shaped upward ferroelectric domains are sketched by the red dashed line. **i** Selected phase profile for different ferroelectric domains as sketched by the white dashed line in (**h**). A phase contrast of 180° is observed between +LP and -LP.

in Fig. 1c, where distinct convex and concave regions exist with opposite curvatures. Since the out-of-plane strain gradients are inversely proportional to the radius of curvature[40], their distribution is then directly quantified from $z(x)$:

$$\kappa_t(x) = \frac{z''(x)}{[1+z'(x)^2]^{\frac{3}{2}}} = \frac{\frac{A}{\sigma^2}\exp\left[-(x-\mu)^2/2\sigma^2\right]\left(1-\frac{(x-\mu)^2}{\sigma^2}\right)}{\left[1+A^2\frac{(x-\mu)^2}{\sigma^4}\exp\left[-(x-\mu)^2/\sigma^2\right]\right]^{\frac{3}{2}}}, \quad (2)$$

where the value and sign of $\kappa_t(x)$ represent the magnitude and direction of the strain gradients, respectively. Figure 1d displays the inhomogeneous distribution of the strain gradients. Because the flexoelectric field is proportional to the strain gradient, the spatial distribution of $\kappa_t(x)$ indicates that positive and negative flexoelectric fields emerge on a scale of several tens of nanometers, which are fundamentally allowed to reverse the polarizations in two-way.

## Experimental realization of two-way flexoelectric control

To validate the strategy of two-way flexoelectric control, ultrathin ferroelectric CIPS was studied in this work. The crystal structure of CIPS can be described by a sulfur framework with metal cations (Cu and In) and P − P pairs filling octahedral voids (see Fig. 2a)[41,42]. The Cu cations can occupy intralayer or interlayer positions, which leads to inversion symmetry breaking and results in low-polarization and high-polarization states in the out-of-plane direction[43]. The low-polarization and high-polarization states can be interconverted by applying an electric field[44-47]. For simplification, we use ±LP and ±HP to denote the four polarization states, with the symbol ± indicating the upward and downward orientations, respectively, in the out-of-plane direction. Within each layer, the Cu ions have three equivalence sites, as indicated by the red dashed line in Fig. 2a, indicating threefold rotational symmetry[48].

The ferroelectricity of the ultrathin CIPS was verified by piezo-response force microscopy (PFM) under ambient conditions, as shown in Fig. 2b, c. The CIPS flakes were mechanically exfoliated from the bulk material and transferred onto polymer or gold-coated silicon substrates via the all-dry transfer technique (detailed in the Methods section). The characteristic Raman modes of CIPS with a thickness of 22 nm were observed at 275, 325, and 384 cm$^{-1}$, which confirmed its single-crystalline nature and are consistent with previous studies[49] (see Fig. S6). The spontaneous ferroelectric domains were visualized by PFM (see Fig. 2b), indicating the high sample quality of CIPS[43,50]. To further verify the ferroelectricity, a typical butterfly-like ferroelectric hysteresis loop was measured in the PFM phase spectra (see Fig. 2c). The coercive electric field ($E_c$) for polarization switching is found to be ~$3 \times 10^5$ V/cm.

Figure 2d–i summarizes our main results for the flexoelectric modulation of the polarizations. In the measurements, we utilized 4% poly(methyl methacrylate) (PMMA) (see Methods) as a flexible substrate. The PMMA substrate helps shape the deformation in the CIPS after the tip is retracted, allowing quantitative analysis of the tip-induced deformation and the corresponding flexoelectric field. As depicted in Fig. 2d and g, circular holes with a depth of 12 nm appeared after tip imprinting. A downward dot-like nanodomain and an upward ring-shaped nanodomain were generated accordingly, as visualized by the PFM phase images in Fig. 2e and h. In particular, for unconventional polarization switching in the ring-shaped domain, a nearly 180° PFM phase contrast was observed between adjacent domains (see Fig. 2i), further confirming the reversed ordering of the polarizations.

The observed two-way flexoelectric control in the CIPS can be quantitatively interpreted with the abovementioned model of soft deformation. By analyzing the cross-sectional line profile of the tip-induced indentation in the CIPS, we found that the sample height change is well depicted by Eq. (1) (see Fig. 2f). The corresponding strain gradient distribution $\kappa_t(x)$ is subsequently calculated by Eq. (2) and is presented in Fig. S7, which shows three regions with positive, negative, and positive values on a scale of ~50 nm. The maximum strain gradients reach $0.7 \times 10^7$ m$^{-1}$ and $-1.7 \times 10^7$ m$^{-1}$ in the upward and downward directions, respectively. Since the flexoelectric field is proportional to the strain gradient, it is expressed as

$$E_{flexo} = \frac{1}{\varepsilon_r \varepsilon_0}(\mu_{ijkl}\kappa_t(x)), \qquad (3)$$

where $\varepsilon_0$ is the permittivity of free space, $\varepsilon_r$ is the relative dielectric permittivity, and $\mu_{ijkl}$ is the flexoelectric coefficient (a fourth-rank polar tensor). For CIPS at room temperature, $\varepsilon_r$ is ~40, and the mesoscopic flexoelectric coefficient $\mu$ is simplified to ~10.8 nC/m[16]. Therefore, the maximum flexoelectric fields ($E_{flexo}$) in the indentation region are $2.13 \times 10^6$ V/cm and $-5.18 \times 10^6$ V/cm in the two directions, which are much greater than the $E_c$ in the CIPS and thus are sufficient to switch the polarizations. As a result, a downward nanodot domain was generated in the upwardly pre-polarized samples, while an unconventional ring-shaped nanodomain with upward polarization was observed on the contrary.

## Effect of the tip force on domain control

To further quantify the tip-induced unconventional flexoelectric effect in ultrathin ferroelectrics, we next investigated the impact of various imprinting forces on domain switching and structure. As depicted in Fig. 3a, the indentation depth and area in the CIPS can be linearly controlled with a gradually increasing tip loading force. Accordingly, the upward strain gradients are significantly tuned from $0.6 \times 10^7$ m$^{-1}$ – $1.4 \times 10^7$ m$^{-1}$ (see Fig. 3b). As the depth reached 20 nm, we noticed a significant change in the sample morphology with the appearance of a convex feature around the indentation,

which was attributed to the tip-induced outward squeezing of the PMMA. To avoid such surface bumps, the depth was kept under 20 nm in the study. Within this range, we observed two other distinct ferroelectric domain structures in addition to the ring-shaped domains.

Under a weak loading force with an indentation depth <5 nm, a tri-petal upward domain was observed, as depicted in Fig. 3c and Fig. S12. The emergence of the tri-petal domain in CIPS is the manifestation of lattice symmetry and anisotropy at the macroscopic scale. For CIPS, first-principles calculations indicate that equivalent deformations along the <100> and <1–10> crystal axes have differing total energies in stable structures, indicating anisotropic flexibility along different crystal axes (see Fig. 3f and Supplementary Note 2)[51–55]. In other words, under the same imprinting force, larger bending and flexoelectric fields are expected along the in-plane <1–10> direction. Therefore, the tip-induced spherical indentation in CIPS is modified by its anisotropic flexibility, resulting in a tri-petal domain pattern near the threshold of flexoelectric polarization reversal. With increasing depth, the domains gradually evolve from tri-petal structures to ring shapes.

As the imprinting force further increases, an anomalous ferroelectric region emerges with enhanced phase contrast in the ring domain center (see Fig. 3d). By examining the line profile in the phase image, we found that the difference in the PFM phase contrast between adjacent domains in Fig. 3e strikingly differs from that in Fig. 2i. The corresponding phase of the new ferroelectric state is different from that for upward polarization and shows a nearly 300° difference from the neighboring downward polarization. Since the sign of the central strain gradient $\kappa_t(x)$ remains negative when the tip force is increased, the observed phase anomaly possibly indicates the generation of a downward -HP due to the interlayer migration of Cu ions (see Fig. 3g). In CIPS, the piezoelectric coefficients of the LP and HP states are negative and positive, respectively[50]. The same stimulating voltage leads to opposite piezoelectric responses in these two states. As shown in Fig. S13, the phase hysteresis loop from the HP is opposite to the anti-clockwise case in the LP state, which originates from the different phase lags in the PFM measurements. Therefore, the observed phase anomaly is possibly due to add-on of the phase lag difference (~180°) in single PFM phase imaging when encountering the transition from +LP to -HP or vice versa. Additionally, in the ring center, the downward $E_{flexo}$ is calculated to be $10.2 \times 10^6$ V/cm (see Fig. S13), which is comparable to the theoretical field threshold (~$10^7$ V/cm)[56] for inducing the transition of high-polarization states.

## Intact flexoelectric domain engineering

With this information on the quantitative characteristics of two-way flexoelectric control in ultrathin CIPS, we finally demonstrated a shape-free mechanical polarization switch and domain engineering at the nanoscale. To avoid shaping, the concentration of the flexible substrate PMMA was increased to 10%, which provides enough elastics. On a 10% PMMA substrate, the tip-induced indentation cannot be permanently shaped. This feature benefits the use of tip-induced flexoelectric control as a transient operation for polarization switching while leaving the sample noninvasive. Figure 4a summarizes the procedure of noninvasive flexoelectric control in ultrathin CIPS. With the imprinting of an AFM tip, the sample is deformed slightly within the elastic limit. The ferroelectric polarizations are accordingly switched due to the flexoelectric effect. After that, the AFM tip is retracted. Due to the elastics of the substrate, the morphology of the CIPS can recover to its original state, while the switched polarizations are preserved. This nanoscale flexoelectric control can be repeatedly implemented, allowing the realization of high-density ferroelectric nanodomains and purely mechanical writing/erasing on domains.

Figure 4b−g shows the experimental realization of intact flexoelectric control of the domains. In the spontaneous upward domain from a sample with thickness of 32 nm, transient tip imprinting was

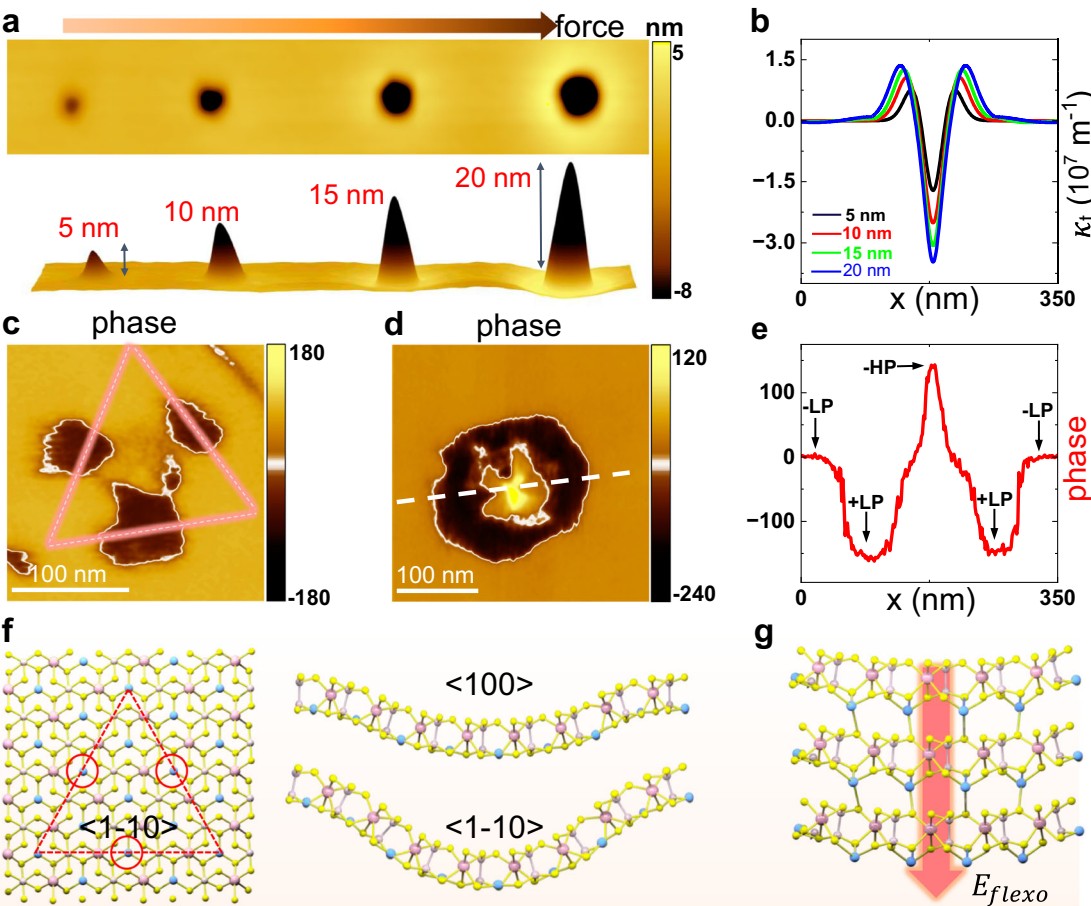

**Fig. 3 | The effect of loading forces on polarizations and domains. a** Topography image of tip-induced indentations with different loading forces from 3–6 µN. **b** Corresponding strain gradients of the different indentations in (**a**). **c** PFM phase image of a CIPS flake at an indentation depth of 5 nm. The red dashed line indicates the in-plane anisotropic flexibility. **d** PFM phase image of a CIPS flake at an indentation depth of 17 nm. **e** Selected phase profile as sketched by the white dashed line in (**d**). A phase anomaly with a contrast of 300° indicates the emergence of -HP states in the CIPS. **f** Schematic of the trigonal symmetry of the CIPS lattice (left panel) and schematic of anisotropic flexibility along the <100> and <1–10> crystal axes (right panel). **g** Crystal structure of the -HP state in CIPS. The Cu ions migrate into the vdW gaps between layers under the tip-induced flexoelectric field.

repeatedly implemented at different locations with a loading force of 4 µN, as indicated by the white dotted line (see Fig. 4d). The sample morphology was recorded before and after the imprinting process. As illustrated in Fig. 4b, c, we found a negligible change in the sample morphology after retracting the AFM tip. The statistics on height variation are ~1 nm for both cases, which indicates the restoration of the original morphology in ultrathin CIPS after tip pressing (more details can be found in Fig. S16). In contrast, a 3 × 3 array of ferroelectric nanodomains was clearly visualized in the PFM phase image (see Fig. 4e). In this case, the ferroelectric polarizations within the region are switched to a downward orientation with the generation of spatially separated nanodomains with diameters as small as ~80 nm. The density of nanodomains can therefore be well controlled by setting the lateral and longitudinal steps in AMF tip imprinting. With a 150 nm step resolution, we realized a high density of nanodomains in 1 µm², equal to an available storage density of 31.4 Gbit/in² (see Fig. S14). Similarly, as indicated in Fig. 4f, g, a 4 × 4 array of upward ferroelectric nanodomains was selectively generated within a spontaneous downward domain.

By changing the step size of the nano-tip imprinting, we next show the two-way mechanical writing/erasing of ferroelectric domains. As depicted in Fig. 5a, with a sufficiently small step size, the mechanically generated nanodomains will partially overlap between neighbors. By continuous imprinting, arbitrary domain

structures at large scales, such as stripes or squares, can be mechanically realized. For simplicity, we showed the mechanical writing of an unconventional upward domain. By employing a tip imprinting step size of 20 nm along the x-axis and 200 nm along the y-axis, separated stripe ferroelectric domains were artificially written in a spontaneous domain (see Fig. 5b, c). By further reducing the step to 20 nm along the y-axis, a square ferroelectric domain with upward polarization was achieved, as depicted in Fig. 5d, e. In addition, we further showed that the mechanically engineered domains can be re-polarized by an external electric field (see Fig. S18). In general, our experiment suggested that arbitrary flexoelectric control might be universal for 2D ferroelectric materials, which can stimulate further widespread studies in other ultrathin ferroelectrics, such as freestanding perovskite oxides.

## Discussion

We propose and verify a mechanical approach for achieving arbitrary control of ferroelectric polarization in 2D ferroelectrics at the nanoscale. Our approach enables noninvasive control of ferroelectricity in ultrathin ferroelectrics, facilitating the use of mechanical methods as an efficient alternative to reliably switch ferroelectric polarizations at the 2D limit. In particular, for ultrathin ferroelectric CIPS, in-plane anisotropic flexibility plays a significant role in determining the electromechanical coupling behaviors, and unique

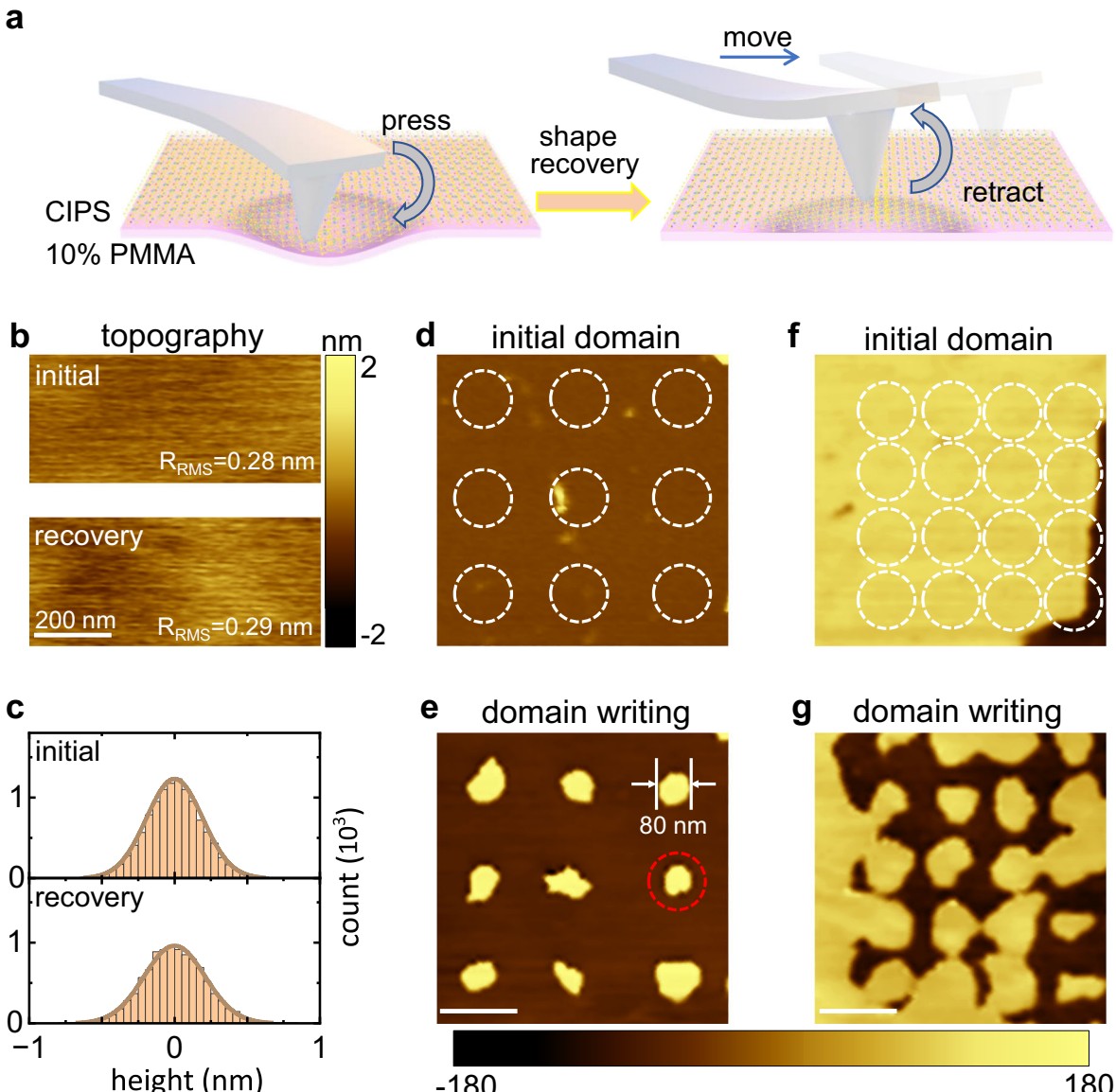

**Fig. 4 | Transient flexoelectric control in ultrathin CuInP₂S₆. a** Flow diagram of transient flexoelectric control in the CIPS. The sample is deformed slightly by tip imprinting within the elastic limit. The ferroelectric polarization is switched via the flexoelectric effect. By retracting the tip, the sample morphology recovers, and the switched ferroelectric polarization is preserved. **b** Topography images before (upper panel) and after (lower panel) the transient flexoelectric control operation. **c** Corresponding statistical histograms of the height variations in (**b**). **d**, **e** Initial and written PFM phase images in a spontaneous upward domain. The white dashed circles in (**d**) indicate the imprinting area of each tip. The generated downward nanodomain is highlighted with a red dashed circle. The scale bar is 200 nm. **f**, **g** Initial and written PFM phase images in a spontaneous downward domain. The scale bar is 200 nm.

HP ferroelectric states can be experimentally realized via a giant flexoelectric field. Our results suggest that flexoelectric control is efficient at switching complex ferroelectricity in CIPS and other ultrathin ferroelectrics, providing a new tool for exploring exotic polarization-related functionalities. The achieved large strain gradient might offer the opportunity to further study mechanical control of the electronic, optical, and magnetic properties of 2D materials, and can be applied to fabricate controllable bulk photovoltaic devices with extraordinary performance.

## Methods
### Substrate and sample preparations
The flexible substrate was made by spin coating PMMA onto a silicon substrate, as shown in Fig. S5. First, 5 nm titanium (Ti) and 30 nm gold (Au) were evaporated onto the flat silicon substrate via e-beam evaporation to enhance the adhesion of PMMA. PMMA solutions with different concentrations (4%/10%) were spin-coated (6000 rpm/8000 rpm at 40 s) onto the gold-coated silicon substrate, followed by heating on a hot plate at 160 °C for 30 s. Finally, 10 nm of Au was evaporated on the surface of the PMMA as a bottom electrode to ensure good performance during the PFM measurements. The bulk CIPS crystal used in this study was purchased from Onway. The CIPS flakes were obtained by mechanical exfoliation and subsequently transferred onto a flexible substrate via the all-dry transfer technique.

### Raman spectroscopy characterization and PFM measurements
Raman spectroscopy was carried out on a Horiba micro-Raman system (LabRAM HR Evolution) with a 100 X objective lens ($NA = 0.9$) to verify the single-crystalline nature of the CIPS used in our experiment. The excitation wavelength was 633 nm with on-sample power at 150 μW. PFM measurements were performed with a commercial

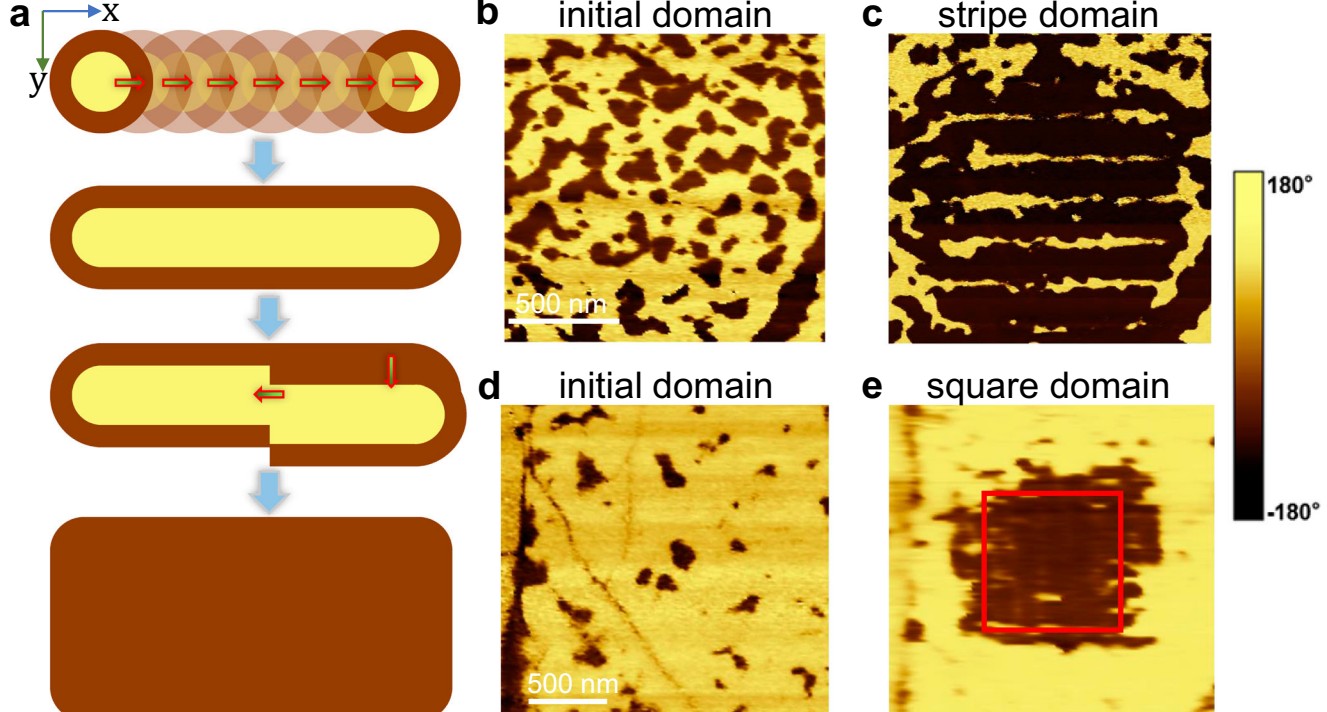

**Fig. 5 | Arbitrary flexoelectric domain engineering. a** Flow diagram of arbitrary flexoelectric domain engineering with nano-tip imprinting technique. **b**, **c** Demonstration of upward stripe domain writing. In a spontaneous domain, by imprinting points with a step size of 20 nm along the *x*-axis and a step size of 200 nm along the *y*-axis, upward stripe ferroelectric domains are obtained. **d**, **e** Demonstration of upward square domain writing. In a spontaneous domain, by imprinting points with a step size of 20 nm along the *x*-axis and a step size of 20 nm along the *y*-axis, an upward square ferroelectric domain is mechanically written.

atomic force microscope (AIST-Smart SPM system) under ambient conditions. PFM measurements were performed by using a Pt/Ir-coated soft tip (Nano World SCM-PIT-75) with a spring constant of 3 N/m and applying an AC voltage ($V_{AC} = 2\,V$) under a tip-sample contact resonant frequency (~290 kHz). We used grounded metallic contact connected to the bottom electrode to screen the local electrostatic field.

## Data availability

The data that support the findings of this study are available from the corresponding author upon reasonable request.

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

## Acknowledgements

This work is supported by the CAS Project for Young Scientists in Basic Research (Grant No. YSBR-049), the Innovation Program for Quantum Science and Technology (Grant No. 2021ZD0302800), the National Key Research and Development Program of China (Grant No. 2018YFA0306600), the Fundamental Research Funds for the Central Universities (Grant No. WK3510000013), and the Anhui Initiative in Quantum Information Technologies (Grant No. AHY170000). This work was partially carried out at the USTC Center for Micro and Nanoscale Research and Fabrication.

## Author contributions

H. Z. conceived the idea and supervised the research. H. L., Q. L., and J. F. prepared the samples, fabricated the devices, and carried out the measurements. H. L., S. Z., Z. F. and H. Z. analyzed the data and wrote the paper. All the authors commented on the manuscript.

## Competing interests
The authors declare no competing interests.
