## [Peer Review File · Nature Communications]

Reversible flexoelectric domain engineering at the nanoscale in van der Waals ferroelectricsEditorial Note: Parts of this Peer Review File have been redacted as indicated to remove third-party material where no permission to publish could be obtained.

REVIEWER COMMENTS

Reviewer #1 (Remarks to the Author):

This work reports that ferroelectric polarization and domain structures can be tuned in two-dimensional CuInP_2S_6 via the nano-tip imprinting technique. The bidirectional flexoelectric control is attributed to the extended tip-induced deformation in two-dimensional systems with innate flexibility at the atomic scale. By employing an elastic substrate, artificial ferroelectric nanodomains with lateral sizes as small as ~ 80 nm are noninvasively generated in an area of $1 \mu\text{m}^2$, equal to a density of 31.4 Gbit/in². However, there are still some concerns about this work. Therefore, I think this manuscript can be considered for publication in Nature Communications after a major revision. The following are some main comments.

(1) For the simulations in Fig. S1, it is unclear how the Cu is distributed in the three different models. For Fig. S1e, the authors apply the In atoms to indicate the bending of the CIPS. However, I think the bending may not significantly change during the relaxations in the model, which should be close to the initial bending models.

(2) Based on the mathematic models of the authors, the proposed strain models in two directions should exist in all ultra-thin materials. Compared to the real models, I do not find a very obvious upward curvature as shown in Fig. S1f.

(3) It is unclear why CIPS was chosen as the candidate to realize the two-way flexoelectric control of the materials. Are there any requirements for the materials to realize such a phenomenon? More discussions should be supplied from this perspective.

(4) After the retraction of the AFM tip, the sample morphology shows quite evident decreases in the intensity as shown in Figure 4c. How to explain such a result? Meanwhile, what about the surface morphology after repeated retraction of the AFM tip in the materials?

(5) Since the authors have constructed the bending structures of the materials. It is interesting to show how the bending modulates the electronic structures of different elements, especially at the upward and downward strain areas.

(6) As shown in Figure 5, the corresponding pattern of the ferroelectric domains is quite rough. Are there any further techniques that can be applied to improve the overall control accuracy of the domain?

(7) I have noticed previous research by the authors about large-scale domain engineering in CIPS (Nano Lett. 2022, 22, 8, 3275–3282). What are the key advances of current work when compared to this one, which also seems to apply the strain control of the ultrathin structures of CIPS?

(8) Are there any possible applications in devices based on the proposed two-way control of ferroelectric domains? More perspectives and discussions should be supplied.

Reviewer #2 (Remarks to the Author):

The paper presents a novel approach in the field of ferroelectric materials by demonstrating the reversible switching of polarization in 2D CIPS through mechanical force. The process enables the creation of artificial ferroelectric nanodomains with high density.

The study showcases three new aspects: the use of an elastic substrate to exploit large strain gradients from an AFM probe, the keen observation of bidirectional curvature in the induced strain, and the innovative utilization of the upward curvature to achieve reversible regulation of polarization. These novel strategies distinguish the work from previous studies that relied on free-standing films over a hole, and open new possibilities in data storage and flexoelectronics.

I recommend it published in Nature Communications and have several minor comments:

1. The authors mentioned that the depth of deformation is more reliable for illustrating the flexoelectric field induced by the probe, but I still suggest them provide the information of applied force, especially for the study based on 10% PPMA without deformation, which helps readers to repeat these experiments.
2. Can those flexoelectric-engineered domains be polarized again by an electric field?
3. Since the sample's thickness has a vital influence on the result, did they succeed in other thicknesses except for 22 nm?

Table of Contents

1. Point-by-point response to review report #1 (pages 1 - 10)
2. Point-by-point response to review report #2 (pages 11 - 13)
3. Major changes to the manuscript (pages 14 - 15).

Point-by-point response to Review Report #1

We thank Referee #1 for his/her positive comments that “Therefore, I think this manuscript can be considered for publication in Nature Communications after a major revision.” Here, we have addressed all the issues raised in this review report. A summary of the major changes can be found in the last section of this file.

Question 1: For the simulations in Fig. S1, it is unclear how the Cu is distributed in the three different models. For Fig. S1e, the authors apply the In atoms to indicate the bending of the CIPS. However, I think the bending may not significantly change during the relaxations in the model, which should be close to the initial bending models.

Response: We thank the referee for this question. We apologize for the misunderstanding caused, which is due to the ambiguity of our previous figure caption in Fig. S1, particularly in Fig. S1e.

Figs. S1a-S1d are used to determine which distribution of Cu ions is energetically preferred in the same bending structure. According to the models in Fig. S1a-S1d, the tips have the same imprinting depth. Therefore, the CIPS lattices in Fig. S1a-S1d have the same bending degree. However, in these models, after relaxation, the distributions of Cu ions are different. By comparing their energies, we determined the energetically preferred distribution of Cu ions. The structure in Fig. S1d has a lower energy than those in Fig. S1a-S1c, suggesting that nano-tip imprinting will lead to a certain polarization direction with Cu ion migration under the tip. These calculated results originate from the flexoelectric effect of CIPS.

To clarify this issue, we have revised the caption of Fig. S1 in the revised supporting information, which is provided here for the referee: “**Figure S1. a** Bending configuration of CIPS caused by nanotip imprinting. b-d Three different distributions of Cu ions in the CIPS lattice with the same bending degree. In all three configurations, the positions of all the atoms are relaxed. By comparing their energies, it was found that the flexoelectric effect leads to a certain polarization of the CIPS under tip imprinting. The energy for placing all the Cu ions downward corresponds to zero potential energy. e The geometric shape of the bending CIPS. For simplicity, the bended CIPS here is represented by the positions of the relaxed In atoms. The deformation is well fitted by a Gauss-type function. From the second derivatives of the Gaussian fitting, we find inflection regions of convex and concave curvatures, as indicated by the blue dashed lines and arrows. f The calculated two-way curvatures in the bent CIPS. The

positive and negative values denote upward and downward strain gradients, respectively."

Question 2: Based on the mathematic models of the authors, the proposed strain models in two directions should exist in all ultra-thin materials. Compared to the real models, I do not find a very obvious upward curvature as shown in Fig. S1f.

Response: We thank the referee for this comment. To introduce two-way flexoelectric control in ultrathin materials, a key of the main finding in our study is the extended deformation caused by the nano-tip. The extended deformation requires that the material itself possess remarkable flexibility and superelasticity, which has been reported in recent studies on free-standing perovskite oxides and vdW 2D materials (see *ACS Nano* 2020, 14, 5053–5060; *Science* 2019, 366, 475–479; *Nano Lett.* 2014, 14, 5097–5103; and *Nat. Commun.* 2017, 8, 15815). To the author's knowledge, flexibility has been recognized as a distinct feature of vdW 2D materials. Therefore, we believe that the proposed strategy for two-way flexoelectric control might be a universal route to mechanically modulate polarization in ultrathin vdW 2D ferroelectrics but might not be applicable to all ultrathin materials.

Mathematically, the tip-induced deformation in ultrathin 2D ferroelectrics is modeled by the following formula:

$$z(x) = z_0 + Ae^{-\frac{(x-\mu)^2}{2\sigma^2}},$$

where $z(x)$ represents the height change of the surface morphology and z_0 and A are constants. The validity of using the Gaussian curvature function has been proven in our experimental measurements, which show perfect fitting results (see Fig. 2f). As the referee mentioned, upward bending might not be clearly observed in the geometric shape of the bending CIPS, as shown in Fig. R1a (used as Fig. S1e in the revised supporting information). However, by applying an additional second derivative to the above Gaussian model, the strain gradients can be directly quantified via

$$\kappa_t(x) = \frac{z''(x)}{[1+z'(x)^2]^{\frac{3}{2}}},$$

where the value and sign of $\kappa_t(x)$ represent the magnitude and direction of the strain gradients, respectively. The spatial distribution of $\kappa_t(x)$, as shown in Fig. R1b (used as Fig. S1e in the revised supporting information), shows positive and negative values in different regions. Since the strain gradient is mainly determined by the radius of

curvature, the positive and negative values of $\kappa_t(x)$ clearly indicate upward and downward curvatures, respectively. To highlight the upward bending and curvature in our model, we add dashed lines and arrows to Fig. S1e to help enhance the clarity. In addition, we have revised the captions of Fig. S1e and S1f to elucidate the existence of upward curvature in the model. The revised caption is given here for the referee: “..... **e** The geometric shape of the bending CIPS. For simplicity, the bended CIPS here is represented by the positions of the relaxed In atoms. The deformation is well fitted by a Gauss-type function. From the second derivatives of the Gaussian fitting, we find inflection regions of convex and concave curvatures, as indicated by the blue dashed lines and arrows. **f** The calculated two-way curvatures in the bent CIPS. The positive and negative values denote upward and downward strain gradients, respectively.”.

Figure R1. **a** The geometric shape of the bending CIPS. For simplicity, the bended CIPS here is represented by the positions of the relaxed In atoms. The deformation is well fitted by a Gauss-type function. From the second derivatives of the Gaussian fitting, we find inflection regions of convex and concave curvatures, as indicated by the blue dashed lines and arrows. **b** The calculated two-way curvatures in the bent CIPS. The positive and negative values denote upward and downward strain gradients, respectively.

Question 3: It is unclear why CIPS was chosen as the candidate to realize the two-way flexoelectric control of the materials. Are there any requirements for the materials to realize such a phenomenon? More discussions should be supplied from this perspective.

Response: We appreciate the reviewer’s concern about the material choice. The demonstration with 2D CIPS is motivated for the following reasons.

First, in the 2D community of ultrathin ferroelectrics, CIPS is one of the most representative ferroelectric materials and has stimulated worldwide research efforts. The out-of-plane ferroelectricity in CIPS originates from the migration of Cu ions, which is distinct from other 2D ferroelectric materials such as α - In_2Se_3 and recently emerged sliding ferroelectrics. Furthermore, CIPS was first used to study the flexoelectric effect in 2D ferroelectrics (see our previous study in *Nano Lett.* 2022, 22, 3275–3282 and Li's study in *Sci. Adv.* 2023,14, 2, 379-386). To maintain the consistency of our research on this topic, we used the CIPS in this study.

Second, the migration of Cu ions results in a unique interplay between ferroelectric polarization and ionic conductivity in CIPS. This coupling behavior reinforces the challenge of reliable polarization switching in ultrathin CIPS with external voltages. An advanced strategy with greater effectiveness in polarization control is highly needed for ultrathin CIPS.

Third, featuring ionic migration, CIPS shows unique high- and low-polarization states. The high-polarization states promise to provide more applications in ferroelectric devices. However, at the current stage, it is challenging to achieve a stable high-polarization state. Theoretical calculations indicate that the threshold electric field required to induce a high-polarization state is 1×10^7 V/cm, which is extremely difficult to realize experimentally. The inherent flexoelectric field applied by the tip imprinting may result in a stable high-polarization state in CIPS.

In general, we believe that the demonstrated two-way flexoelectric control does not impose specific requirements on ultrathin ferroelectric materials. The only requirement is that the material should have excellent mechanical properties to ensure the application of a sufficient strain gradient. To date, excellent flexibility has been found in many vdW 2D materials. Therefore, we believe that the demonstrated two-way flexoelectric control might be universal for ultrathin 2D ferroelectrics. As suggested by the referee, we have added more discussion to the revised manuscript.

Question 4: After the retraction of the AFM tip, the sample morphology shows quite evident decreases in the intensity as shown in Figure 4c. How to explain such a result? Meanwhile, what about the surface morphology after repeated retraction of the AFM tip in the materials?

Response: We thank the referee for the careful review. The intensity in Fig. 4c represents the distribution of the sample morphology height. The decreases in the intensity, as well as the broadening of the statistical distribution curve, indicate an increase in the sample surface roughness. In our study, the initial sample surface

roughness was found to be within 1 nm. After the imprinting and retraction of the AFM tip, a very slight change in roughness at ~ 0.1 nm is observed, as shown in Fig. 4c. This result is within the measurement accuracy of our AFM.

To further evaluate the increase in surface roughness caused by the tip imprinting technique, we performed a series of repeated imprinting experiments (1 to 15 cycles) on the same sample area. As shown in Fig. R2b and R2c (used as Fig. S16 in the revised supporting information), the morphological roughness slightly increases with the number of cycles. These results indicate that the demonstrated transient two-way flexoelectric control is noninvasive and safe for maintaining the sample morphology.

Figure R2. **a** Histograms of the topography before and after the operation. **b,c** The AFM topography and corresponding roughness of repeated imprinting areas from 1 to 15 cycles.

Question 5: Since the authors have constructed the bending structures of the materials. It is interesting to show how the bending modulates the electronic structures of different elements, especially at the upward and downward strain areas.

Response: We thank the referee for this suggestion. Following this suggestion, we have calculated the projected density of states (PDOS) of Cu ions in bended CIPS in the revised manuscript. We compare the PDOS of Cu ions in the flat and curved models. The results have been added to the revised supporting information. For the convenience of the review, we provide these results and discussions here for the referee in the following.

The discussion is “The effects of curving on the electronic structures are also studied.

We focus on the Cu ions in the upward and downward strain areas, where the value of the curvature is the largest. Therefore, it is natural that the electronic structures of these Cu ions could significantly change. For comparison, the PDOSs of Cu in flat and curved CIPs were calculated (as shown in Fig. R3a and R3b, respectively). The dz^2 orbital of Cu significantly changes. In the flat model, the PDOS of the dz^2 orbital (red curve) is characterized mainly by multiple peaks. In contrast, in the curved model, there is only one main peak, which becomes sharper than that in the flat model. At the same time, other peaks shrink remarkably or vanish. These results indicate that the curving causes the dz^2 orbital of Cu in the strain area to be more localized. This change in the electronic structure is understandable. The spatial orientation of the dz^2 orbital is out-of-plane and perpendicular to the CIPs film. Therefore, in the curved model, the out-of-plane bending strain (upward and downward) has a large effect on the dz^2 orbital. Furthermore, the localization of dz^2 orbitals can be attributed to the additional polar displacement of Cu ions along the out-of-plane direction due to the flexoelectric effect in the curve model.”.

Figure R3. **a** PDOS of Cu in a flat CIPs. **b** PDOS of Cu in a curved CIPs $\langle 100 \rangle$.

Question 6: As shown in Figure 5, the corresponding pattern of the ferroelectric domains is quite rough. Are there any further techniques that can be applied to improve the overall control accuracy of the domain?

Response: We thank the referee for this question. For the out-of-shape domain pattern in Figure 5, a potential reason lies in the coupling between the ferroelectric polarizations and the migration of Cu ions in CIPs. The migration of Cu ions strongly affects the patterns of artificial domains (see Fig. S18).

From a technical perspective, the control accuracy of intact flexoelectric domain

engineering can be improved by tuning the flexibility or elasticity of the substrate. For example, on a hard Si substrate, a near-perfect square domain with downward polarization can be mechanically written in CIPS, as shown in Fig. R4a. This result suggests that if the hardness or elasticity of the substrate used in our study can be increased, the transient control accuracy can be improved. An intuitive understanding might be that the decrease in the flexibility of the substrate results in a reduced inconsistency in transferring the mechanical modulation of polarization. Therefore, we prepared substrates with different hardnesses by changing the thickness of Au (10 nm and 20 nm) on 10% PMMA, as shown in Fig. R4b-R4c. In the experiments, we perform tip imprinting in the rectangular areas, as indicated by the red dashed lines in Fig. R4b-R4c. By comparing the results, we find that transient tip imprinting on a 20 nm Au/10% PMMA substrate is more efficient and more accurate. The mechanical written domains on the 20 nm Au/10% PMMA substrate are closer to the predefined rectangular shape than the domains on the 10 nm Au/10% PMMA substrate. These findings highlight the significant influence of substrate flexibility on the mechanical manipulation of ferroelectric domains. To further optimize our technique, future explorations on suitable polymer substrates are needed.

Figure R4. a-c The PFM phase images of tip imprinting induced domain structures on Si substrate, 10 nm Au/10% PMMA, and 20 nm Au/10% PMMA. The tip imprinted areas are indicated by red dashed lines, showing downward polarization writing.

Question 7: I have noticed previous research by the authors about large-scale domain engineering in CIPS (*Nano Lett.* 2022, 22, 8, 3275–3282). What are the key advances of current work when compared to this one, which also seems to apply the strain control of the ultrathin structures of CIPS?

Response: We thank the referee for this question. In our previous study (*Nano Lett.*

2022, 22, 8, 3275–3282), we aimed to artificially generate a single ferroelectric domain at a large scale, such as a lateral domain at the scale of several hundred microns. Therefore, we prepared a kind of polymer substrate with periodic ripples, which enabled the formation of large-scale striped ferroelectric domains via the fixed flexoelectric effect, as shown in Fig. R5a (reproduced from our previous study, *Nano Lett.* 2022, 22, 8, 3275–3282). However, this approach is highly dependent on the substrate. The artificial domain pattern is pinned by the sharp morphology of the rippled substrate. As shown in Fig. R5b (reproduced from our previous study, *Nano Lett.* 2022, 22, 8, 3275–3282), the hysteresis loops from the downward/upward domains show a large shift to the negative/positive voltage range, indicating that ferroelectric polarization can only be stabilized in one direction in the bending region. As a result, polarization reversal is difficult to realize.

Motivated by the drawbacks of substrate-induced flexoelectric domain engineering, in the present study, we aimed to develop a mechanical technique that can arbitrarily control two-way polarization. As shown in Fig. R5c, the foremost advance of the current study lies in the ability to arbitrarily and bidirectionally modulate the domains in ultrathin 2D ferroelectrics. The developed technique now ensures reliable domain engineering while preserving the surface morphology of the sample noninvasively, which is significant in contrast to our previous study. In addition, the demonstrated nanotip imprinting technique can realize a much higher strain gradient at $\sim 3 \times 10^7 \text{ m}^{-1}$. In our previous study of substrate strain engineering, the maximum strain gradient was $\sim 1 \times 10^6 \text{ m}^{-1}$. The achieved large strain gradient might offer the opportunity to further study flexoelectric control of the electronic, optical, and magnetic properties of 2D materials.

[Redacted]

Figure R5. **a** Topography image of a CIPS thin flake on a rippled substrate. **b** PFM hysteresis loops of the upward bending region and the downward bending region. **a** and **b** are reproduced from *Nano Lett.* 2022, 22, 8, 3275–3282). **c**. Flow diagram of transient flexoelectric control in the CIPS. The sample is deformed slightly by tip imprinting within the elastic limit. The ferroelectric polarization is switched via the flexoelectric effect. By retracting the tip, the sample morphology recovers, and the switched ferroelectric polarization is preserved.

Question 8: Are there any possible applications in devices based on the proposed two-way control of ferroelectric domains? More perspectives and discussions should be supplied.

Response: We appreciate the suggestion to discuss potential applications of our technique. In this study, we employed tip-based imprinting techniques to realize nanoscale strain gradients as high as 10^7 m^{-1} . For potential device applications, we believe that this technique can be applied to fabricate controllable bulk photovoltaic devices with extraordinary performance.

The bulk photovoltaic effect (BPVE) refers to the conversion of light photons into electricity in polar materials without constructing a p-n junction or Schottky barrier. Therefore, BPVE offers a promising route to surpass the thermodynamic Shockley–Queisser limit in conventional solar cells. Recent advances show that BPVE is particularly pronounced at the nanoscale or in low-dimensional systems (see *Nat. Commun.* 2021, 12, 5896; and *Nat. Commun.* 2022, 13, 3237). Therefore, by utilizing our technique, controllable BPVE devices can be fabricated on demand from both ferroelectric and non-ferroelectric 2D materials. The inherent polarization induced by the flexoelectric effect can effectively separate the photogenerated carriers. Furthermore, nanoscale strain gradients can modulate the band structure of 2D materials, allowing broadening of the absorption spectrum (see *Nat. Photon.* 2012, 6, 866–872). Notably, for ferroelectric materials that inherently possess a BPVE, the superposition of the flexoelectric field might result in an enhanced photoelectric response, thereby improving the efficiency of photovoltaic devices. As suggested by the referee, we have added more discussion to the revised manuscript.

Point-by-point response to review report #2

We thank Referee #2 for his/her recommendation of our manuscript with the highly positive comments and that “These novel strategies distinguish the work from previous studies that relied on free-standing films over a hole, and open new possibilities in data storage and flexoelectronics. I recommend it published in Nature Communications”. As detailed below, we have addressed all the issues raised and revised the manuscript accordingly.

Question 1: The authors mentioned that the depth of deformation is more reliable for illustrating the flexoelectric field induced by the probe, but I still suggest them provide the information of applied force, especially for the study based on 10% PPMA without deformation, which helps readers to repeat these experiments.

Response: We thank the referee for this suggestion. Following this suggestion, we have added the loading force used on the 10% PMMA substrate to the revised manuscript and the loading force used on 4% PMMA to the revised supporting information. It should be noted that the sample thickness and the substrate hardness strongly impact the tip deformation in the tip imprinting technique. Therefore, the dependence of the deformation depth and the applied force must be investigated before performing the tip imprinting technique. This is why we mentioned that the depth of deformation is more reliable for illustrating the flexoelectric field induced by the probe in the manuscript. As suggested by the referee, we show the relationship between the deformation depth and the applied force on the 10 nm Au/4% PMMA substrate for sample thicknesses of 8, 16, 18, and 23 nm in Fig. R6 (used as Fig. S9 in the revised supporting information).

It is clearly found that the tip-induced deformation depth increases proportionally with the applied force for the same sample thickness, while thinner samples exhibit greater deformation when subjected to the same force.

Figure R6. Dependence of the deformation depth and loading force. The tip-induced deformation depth is directly proportional to the applied loading force.

Question 2: Can those flexoelectric-engineered domains be polarized again by an electric field?

Response: We thank the referee for this question. The answer is yes. To test this hypothesis, we checked whether flexoelectric-engineered domains can be polarized again by an external electric field. We first mechanically write a 3×5 array of downward ferroelectric nanodomains in CIPS via repeated tip imprinting with a loading force of $4 \mu\text{N}$. The nanodomains are clearly visualized in the PFM phase image after imprinting, as highlighted by the blue dashed lines in Fig. R7a. Subsequently, we applied a voltage of -5 V via a conductive tip within the mechanically written region. As outlined by the red dashed line in Fig. R7b, the flexoelectric effect engineered domains are clearly erased by the applied electric field, which demonstrates the capability of reconfiguration for these domains. Notably, after applying the external electric field, we observe unexpected domain changes or switches outside of the electric field writing area. This is possibly due to the unexpected migration of Cu ions in CIPS when an external voltage is applied, which results in a change in the ferroelectric domains. Following the reviewer's suggestion, we have added the above results to the revised supporting information.

Figure R7. **a** PFM phase image after tip imprinting with a 3×5 array, where the area is delineated by a blue dashed line. **b** PFM phase image after electric writing at -5 V, where the area is delineated by the red dashed line.

Question 3: Since the sample's thickness has a vital influence on the result, did they succeed in other thicknesses except for 22 nm?

Response: We thank the referee for this question. In this study, we investigated samples with thicknesses ranging from 11 to 32 nm, as shown in Fig. R8 (used as Fig. S17 in the revised supporting information). All of the investigated samples demonstrate effective flexoelectric tunability.

Figure R8. **a-d** PFM phase images after tip imprinting in samples with film thicknesses of 11, 15, 21, and 32 nm.

Major changes to this manuscript

1. We have fixed all the typos in the manuscript. Changes made to the text in the manuscript and the supporting information are marked in red. We have added scale bars to all the figures, if necessary.
2. We have revised the caption of Fig. S1 in the revised supporting information.
3. The calculated electronic structures of Cu in upward-bended CIPS are added to the supplementary information as Supplementary Note 3.
4. The information of applied force in tip printing experiments has been provided in the revised manuscript.
5. In the “Intact flexoelectric domain engineering” section in the main text, we have discussed the surface morphology after repeated imprinting and the possibility that the flexoelectric-engineered domains can be polarized again by an electric field. The sentences are “In addition, we further showed that the mechanically engineered domains can be re-polarized by an external electric field (see Fig.S18).”.
6. In the “Conclusion” section in the main text, we have discussed the possible

applications in devices based on the proposed two-way control of ferroelectric domains. The sentences are “The achieved large strain gradient might offer the opportunity to further study mechanical control of the electronic, optical, and magnetic properties of 2D materials, and can be applied to fabricate controllable bulk photovoltaic devices with extraordinary performance.”.

7. The journal names of the references have been changed to Nature Communications style.

8. A revised supporting information has been prepared, providing more supporting data for the major conclusions in the main text. We outline the major changes in the revised supporting information as follows.

- a) The electronic structures of Cu in the flat and upward-curving CIPS are shown in Fig. S3, which indicates that the curving causes the dz^2 orbital of Cu in the strain area to be more localized. The localization of dz^2 orbitals can be attributed to the additional polar displacement of Cu ions along the out-of-plane direction due to the flexoelectric effect in the curve model.
- b) We add the dependance of deformation depth on loading force for different sample thicknesses in Fig. S9.
- c) We add the sample topography and corresponding roughness after repeated imprinting to Fig. S16.
- d) We add the PFM phase images after intact flexoelectric domain engineering in samples with thicknesses of 11, 15, 21, and 32 nm, as shown in Fig. S17.
- e) Fig. S18 shows that the flexoelectric-engineered domains can be polarized again by an electric field.

REVIEWERS' COMMENTS

Reviewer #1 (Remarks to the Author):

The authors have devoted great efforts to addressing all the comments by supplying additional experimental data and detailed discussions to improve the manuscript during the revision. I think the authors have shown a good attitude during the revisions and I'm satisfied with the revised manuscript, which is suitable for publication now. Therefore, I think this manuscript can be accepted for publication in Nature Communications.

Reviewer #2 (Remarks to the Author):

The author's responses address my questions. Figure 5 may need a scale bar.

Point-by-Point Response

To Reviewer #1

We thank Review #1 for his/her recommendation of our manuscript.

To Reviewer #2

We thank Review #2 for his/her recommendation of our manuscript. A scale bar is added to Figure 5.